# Occurrence of *n*-Alkanes in Vegetable Oils and Their Analytical Determination

**DOI:** 10.3390/foods9111546

**Published:** 2020-10-26

**Authors:** Ana Srbinovska, Chiara Conchione, Luca Menegoz Ursol, Paolo Lucci, Sabrina Moret

**Affiliations:** Department of Agri-Food, Environmental and Animal Sciences, University of Udine, 33100 Udine, Italy; chiara.conchione@uniud.it (C.C.); menegozursol.luca@spes.uniud.it (L.M.U.); paolo.lucci@uniud.it (P.L.); sabrina.moret@uniud.it (S.M.)

**Keywords:** *n*-alkanes, olive oils, vegetable oils, geographical origin, variety differentiation, sample preparation, chromatography, statistical analysis

## Abstract

Vegetable oils contain endogenous linear hydrocarbons, namely *n*-alkanes, ranging from *n*-C_21_ to *n*-C_35,_ with odd chain lengths prevalent. Different vegetable oils, as well as oils of the same type, but of different variety and provenience, show typical *n*-alkane patterns, which could be used as a fingerprint to characterize them. In the first part of this review, data on the occurrence of *n*-alkanes in different vegetable oils (total and predominant *n*-alkanes) are given, with a focus on obtaining information regarding variety and geographical origin. The second part aims to provide the state of the art on available analytical methods for their determination. In particular, a detailed description of the sample preparation protocols and analytical determination is reported, pointing out the main drawbacks of traditional sample preparation and possible solutions to implement the analysis with the aim to shift toward rapid and solvent-sparing methods.

## 1. Introduction

*n*-Alkanes are non-polar, stable and scarcely reactive organic compounds, also named paraffins, which in Latin means “with a scarce chemical affinity” [1]. They consist of linear saturated hydrocarbon chains following the general formula C**_n_**H_2n+2_.

Together with squalene, they represent the main components of the unsaponifiable fraction that represents ~1–2% of the oil weight. While some minor compounds of vegetable oils, such as sterols, tocopherols and terpenic alcohols, have been widely studied because of their importance in the characterization and assessment of the oil quality, endogenous *n*-alkanes have attracted little attention, probably because of their supposed minor role in vegetable oil characterization [2,3]. The *n*-alkane fraction of vegetable oils is mainly composed of hydrocarbons in the range of *n*-C_21_ and *n*-C_33,_ with odd terms prevailing on the even ones [4,5]. 

*n*-Alkanes are widely distributed in both plants and other living organisms. Together with other components of the wax fraction (wax esters, fatty aldehydes, triterpenoids, alkyl and sterol esters), they form the external cuticle of plant epidermis, acting as a moisture barrier [6]. Relatively high amounts of *n*-alkanes with different distributions can be also present in soil. Compared to higher plant *n*-alkanes, soil includes hydrocarbons with a wider molecular range, from *n*-C_21_ to *n*-C_47_ [7].

Regarding their origin, *n*-alkanes in plants are thought to be formed by elongation of a preformed fatty acid, followed by loss of the carboxyl carbon. The predominance of alkanes with an odd carbon number strongly suggests that the fatty acid precursors lose only one carbon during their conversion to alkanes [8,9].

Numerous components of vegetable oils are used to check the quality and to detect frauds, for example, sterols to identify seed oils, triglyceride composition to identify esterified oil and triterpene diols to identify olive pomace oil [10]. A number of papers have demonstrated the discrimination potential of endogenous hydrocarbons. Indeed, the *n*-alkane profile can be used to determine the authenticity of expensive edible oils and whether they are admixed with cheaper oils. Furthermore, it is possible to use the quali-/quantitative *n*-alkane profile to distinguish between crude and refined oils of different plant origin and among oils obtained from different olive cultivars and/or of different provenience [11,12].

*n*-Alkanes in vegetable oils can also originate from contamination with mineral oils, more precisely with saturated mineral oil hydrocarbons (MOSH). Mineral oil aromatic hydrocarbons (MOAH), which may accompany the MOSH, are generally separated from the latter before analytical determination, which is performed via gas chromatography (GC) coupled to a flame ionization detector (FID) [13]. Contamination with mineral oils in vegetable oils can occur along the whole processing chain from the field to the packaging [14]. Different methods have been proposed over the years for the determination of aliphatic hydrocarbons (including both endogenous *n*-alkanes and MOSH) in vegetable oils, the most popular being based on column chromatography followed by GC-FID [15,16,17,18,19] or online high-performance liquid chromatography (HPLC)-GC-FID [20,21,22].

Even if not separated from the endogenous *n*-alkanes, MOSH are easily distinguishable due to the lack of predominance of odd-numbered carbon chains and due to the presence of one or more “humps”, also known as “unresolved complex mixtures” (UCM). These humps consist of several hundred unresolved peaks, including *n*-alkanes, isoalkanes and cycloalkanes, which cannot be separated as single peaks, even when using comprehensive GC. Figure 1a,b show the typical LC-GC traces of two extra virgin olive oils (different cultivars) with no detectable MOSH but different *n*-alkane profile, while Figure 1c shows an olive oil with a typical hump due to MOSH contamination.

The aim of this review was to give an overview of the occurrence of *n*-alkanes in edible oils and the potential/suitability of its determination for oil characterization and quality control purposes. Methods for *n*-alkanes determination were also critically reviewed.

## 2. Occurrence in Vegetable Oils 

### 2.1. Olive Oils

Table 1 summarizes total and predominant *n*-alkanes found by different authors in olive oils of different types, varieties and geographical origins.

Most of the authors reported distinct quali-/quantitative *n*-alkane profiles in oils of different variety and provenience. Mihailova et al. [12] found that the predominant *n*-alkanes in extra virgin olive oil (EVOO) were *n*-C_23_, *n*-C_25_, *n*-C_27_, *n*-C_29_ and *n*-C_31_, but the relative proportion differed depending on the country of origin. Other authors found different predominant *n*-alkanes.

From data reported in Table 1, it can be concluded that total *n*-alkanes of different olive oils varied within a relatively large range, from 13.7 to 176.2 mg/kg. In most of the samples, the predominant *n*-alkane was *n*-C_25_ or *n*-C_29_, while in the rest of the sample prevailed *n*-C_23_ or *n*-C_27_.

As an example, according to Sakouhi et al. [23], the olives of the Tunisian *Maski* variety showed a carbon number predominance for *n*-C_23_, *n*-C_25_ and *n*-C_27_, and more generally, the Tunisian variety was characterized by a greater presence of *n*-C_23_, while the Italian olives had an *n*-alkane profile with a prevalence of *n*-C_29_.

### 2.2. Other Vegetable Oils

Even though only a few authors focused their attention on *n*-alkanes in other vegetable oils, it is well recognized that, by using the *n*-alkane profile and composition, as well as total *n*-alkane, it is possible to distinguish between oils of different plant origins and to detect frauds (admixture of high-priced olive oils with cheaper ones).

Among different edible oils, olive oil stands out for its delicate flavor, stability and reported health benefits. The commercial value of olive oil is considerably higher than that of other vegetable oils such as sunflower and corn oil. The adulteration of olive oil with relatively cheap edible oils is economically attractive. Adulteration of extra virgin olive oil with seed oils, refined oil or olive pomace oil is one of the main concerns of the olive industry, and, in this context, *n*-alkane content and profile could represent an interesting parameter to monitor. As an example, by using principal component analysis (PCA), Webster et al. [5] allowed detected the adulteration of extra virgin olive oil with rapeseed or sunflower oils at a very low percentage of the adulterant (0.5%).

Table 2 summarizes works regarding *n*-alkanes detection in vegetable oils different from olive oils, reporting information regarding the variety, the geographical origin, number of the sample analyzed, and total and major *n*-alkanes.

From the data reported in Table 2, it can be concluded that different vegetable oils have very different quali-/quantitative *n*-alkane profiles. In terms of the total *n*-alkane content, most of the oils showed a lower variability range than olive oils, and, in particular, some nut oils showed the lowest *n*-alkane content, e.g., from 7 to 30 mg/kg for walnut oil (3 samples), from 0.6 to 13.9 mg/kg for hazelnut oil (17 samples), while tomato seed oil showed the highest one (from 106.7 to 347.3 mg/kg).

McGill et al. [24] analyzed several types of vegetable oils, reporting *n-*alkane concentrations ranging from 7 (walnut oil) to 166 mg/kg (sunflower oil). They also reported different profiles with short-chain *n*-alkanes (*n*-C_23_, *n*-C_25_, *n*-C_27_) prevailing in olive oil and long-chain *n*-alkanes (*n*-C_27_, *n*-C_29_, *n*-C_31_) predominating in the rest of the vegetable oils, with different order of prevalence.

Hazelnut oil has a lipidic profile very similar to olive oil, making it difficult to detect the fraudulent addition of small quantities of hazelnut oil in olive oil. The most abundant odd-numbered *n*-alkanes in hazelnut oils are those also present in olive oils, but the content of individual *n-*alkanes in olive oils is in general much higher compared to that in hazelnut oils [29] (Table 1 and Table 2).

Giuffrè [30] found 14 compounds in avocado oil ranging from n-C_21_ to n-C_34_, with n-C_24_, n-C_25_ and n-C_23_ being the most abundant.

Later, Troya et al. [3] demonstrated that oils of different botanical origin have different chromatographic profiles. Analysis was performed using mass-spectrometry (MS) detection. Figure 2 shows the GC-MS profile of six different vegetable oils. For grapeseed (A), peanut (D) and corn (E) oils, the most abundant *n*-alkane was *n*-C_31_, followed by *n*-C_29_ and *n*-C_27_ for grapeseed oil, *n*-C_33_ and *n*-C_29_ for corn oil and *n*-C_29_ and *n*-C_30_ the peanut oil. For both hazelnut (B) and sunflower (F) oils, the most abundant *n*-alkane was *n*-C_29_, followed by *n*-C_31_ and *n*-C_27_. Nevertheless, the two oils can be easily distinguished based on other *n*-alkanes. For olive oil (C) the most abundant *n*-alkane was *n*-C_27_, followed by *n*-C_25_ and *n*-C_29_.

Finally, Giuffrè et al. [1] investigated the *n*-alkane composition in tomato seed oils from three different cultivars grown in Southern Italy. The highest *n*-alkane content was found in *Principe Borghese* variety (347.3 ± 4.04 mg/kg), followed by *Rebelion* (216.3 ± 3.05 mg/kg) and *San Marzano* (106.7 ± 0.58 mg/kg). All the varieties were characterized by the major presence of short-chain *n*-alkanes.

In conclusion, *n*-alkanes quali-/quantitative profiles represent a very good fingerprint to distinguish among different vegetable oils.

### 2.3. Parameters Affecting n-Alkane Content and Qualitative Profile

#### 2.3.1. Influence of the Variety

*n*-Alkane composition of virgin olive oils has been widely investigated by different authors [11,26], who concluded that the *n*-alkane content and profile is variety-dependent and could therefore be used as a purity index to prove mono-varietal origin of EVOOs.

Lanzon et al. [25] reported for Spanish virgin olive oils of different cultivars, a wide range of total *n*-alkane concentrations (from 15.3 to 115.8 mg/kg). 

Guinda et al. [26] analyzed virgin olive oils from different Spanish varieties (*Arbequina, Cornicabra, Empeltre, Hojiblanca, Picual*) and observed different *n*-alkanes contents (See Table 2). The major compounds were *n*-C_27_ in the *Arbequina* and *Picual* varieties, *n*-C_25_ in the *Empeltre* variety and *n*-C_29_ in the *Hojiblanca* and *Cornicabra* varieties. The *Empeltre* variety showed the highest *n*-alkane concentration (72–92 mg/kg), with *n*-C_25_, *n*-C_23_ and *n*-C_29_ as predominant compounds, while the *Picual* variety had the lowest concentration (18–31 mg/kg), with the most abundant *n*-alkanes being *n*-C_27_, *n*-C_29_ and *n*-C_25_.

Koprivnjak et al. [11] studied the *n*-alkane composition of three different cultivars *(Leccino, Buza, Bjelica*) from Croatia, harvested during four consecutive years at three different ripening stages. They used linear discriminant analysis to exploit the *n*-alkane identifying the olive cultivar, and observed that all olive oils, except one *Leccino* sample, received the correct assignment (Figure 3). The authors reported that the main components in the *Bjelica* cultivar were *n*-C_29_, *n*-C_25_ and *n*-C_27_, while for the other two cultivars (*Buza* and *Leccino*), the main components were *n*-C_25_, *n*-C_23_ and *n*-C_24_ (See Table 1).

Nartea [7] underlined how much the variety influences the *n*-alkanes profile in Greek EVOOs, also in terms of total amount. It was found that total *n*-alkanes ranged from 132.6 to 241.5 mg/kg for the *Koroneiki* variety, and from 55.1 to 59.0 mg/kg for the *Manaki* variety. In general, in contrast to *Manaki,* which showed *n-*C_25_ and *n*-C_29_ approximately equally distributed, *Koroneiki* displayed a carbon number prevalence for *n*-C_23_, *n*-C_25_, *n*-C_27_ and *n*-C_29_, with a decreasing trend starting from *n*-C_23_.

#### 2.3.2. Influence of the Geographical Origin

A limited number of studies have evaluated quali-/quantitative *n*-alkane composition as a tool to discriminate oils from different countries [2,5,12].

Webster et al. [5] studied the *n*-alkane profile of 20 authentic extra virgin olive oils and 20 retail olive oils of various origins. The highest amount of total *n*-alkanes from *n-*C_15_ to *n*-C_33_ was found in Greek EVOOs (on average 152.5 mg/kg), with *n*-C_23_ and *n-*C_25_ predominating in the others. Similar results were also observed for retail samples. Italian EVOOs had on average 71.7 mg/kg of total *n*-alkanes, while Spanish EVOOs showed the lowest concentration (on average 43.6 mg/kg). In contrast with Greek samples, which showed a characteristic *n*-alkane profile, Italian and Spanish oils showed different *n*-alkane patterns. The authors found on average 70.8 mg/kg of total *n*-alkanes in the Italian retail samples, which is in accordance with the concentration found in the authentic EVOOs from the same country. The distinction among different geographical origins was carried out by principal component analysis (PCA) which, based on the specific data clusters, highlighted valuable differences between *n*-alkane patterns of various olive oils (Greek, Spanish and Italian) (Figure 4).

Mihailova et al. [12] surveyed the molecular distribution of *n*-alkanes ranging from *n*-C_20_ to *n*-C_30_ from eight different Mediterranean countries through the stable hydrogen and carbon isotope compositions (δ^2^H and δ^13^C of *n*-C_29_) to investigate the relation between these variables and regional environmental conditions and to survey how this information could be used for the determination of the geographical origin. High amounts of short-chain *n*-alkanes (*n*-C_23_, *n*-C_25_) were found in the oils from northern and southern Italy, Greece, Morocco, France and Croatia. On the other hand, the oils from central Italy, Slovenia, Spain, Portugal and northern Greece were dominated by longer-chain *n*-alkanes (*n*-C_27_, *n*-C_29_, *n*-C_31_). 

For the statistical analysis, the oil production countries were divided into two regions: southern and northern Mediterranean, according to latitude, showing significant differences with respect of their carbon and hydrogen isotope compositions. The authors stated that δ^13^C and δ^2^H values of Mediterranean olive oils from different countries are significantly affected by the environmental conditions (mean temperature from August to December and relative humidity) resulting in successful discrimination between oils from the southern and northern regions. Nevertheless, they suggested that the combination of carbon and hydrogen isotope compositions of olive oil *n*-alkane could be a useful tool for the geographical distinction of oils.

The concentration of total *n*-alkanes of Spanish olive oils (52.5–65.1 mg/kg) reported by Bortolomeazzi et al. [2] agreed with the data (40.1 ± 7.1 mg/kg) reported by Mihailova et al. [12].

#### 2.3.3. Influence of Maturation Degree and Presence of Leaves

Differences in quali-/quantitative *n*-alkane profile of extra virgin olive oil can be also due to the olive ripening degree [23,31] or the presence of leaves [28].

When comparing the wax cuticular composition of the Italian *Coratina* cultivar at different maturation stages (green and black olives), Bianchi et al. [31] found different *n*-alkane patterns. In both cases, the predominant *n*-alkanes in the olives were *n*-C_25_, *n*-C_27_ and *n*-C_29_. Green and black olives differed in the presence of a higher amount of *n*-C_29_ (47% of total *n*-alkanes) in green olives and the higher presence of *n*-C_27_ (27% of total *n*-alkanes) in black ones.

A study on avocado pulp oil during fruit ripening showed the changes in *n*-alkane composition [30]. At the first harvesting date (25.20 ± 0.45 mg/kg) the total *n*-alkane content was higher compared with the second (18.41 ± 0.08 mg/kg) and the third (16.77 ± 0.11 mg/kg) harvesting date.

In contrast with other authors, Koprivnjak et al. [11], who analyzed EVOOs of three different varieties (*Bjelica, Buza and Leccino*), did not find significant differences in relation to the period of harvesting (from the middle of October until the end of November). Only *Leccino* samples showed a decreasing trend during maturation, especially for *n*-alkanes with less than thirty carbon atoms.

Sakouhi et al. [23] observed important variation in the content of *n*-alkanes during fruit maturation and found that the major compounds *n*-C_25_, *n*-C_27_ and *n*-C_23_, at the beginning of *Meski* olive development (21st week after the flowering date), were 162.3, 110.2 and 73.6 mg/kg, respectively. From the 21st week after the flowering date, the *n*-alkanes started to decrease gradually until complete maturity of the *Meski* olive (38th week after the flowering date), reaching values of 22.1, 13.1 and 1.04 mg/kg for *n*-C_25_, *n*-C_27_ and *n*-C_23_, respectively.

Mihailova et al. [28] studied the *n*-alkane profiles in cuticular waxes of olives and leaves of different Italian cultivars (*Frantoio, Maraiolo, Leccino*) during ripening and found that at the start of the season (July) the most prevalent *n*-alkanes in olive fruits were those with long carbon chains (*n*-C_29_, *n*-C_31_), while by the end of the season (November) shorter-chain *n*-alkanes (*n*-C_25_, *n*-C_27_, *n*-C_29_) predominated. Different from the olives, for the leaves, no variation in terms of *n*-alkanes distribution during maturation was observed. Leaves are characterized by the presence of higher amounts of long-chain *n*-alkanes (*n*-C_29_, *n*-C_31_ and *n*-C_33_). The same authors reported important differences between *n*-alkane profiles in EVOOs with different leaf content and demonstrated that a higher presence of leaves results in a higher concentration of long-chain *n*-alkanes (*n*-C_31_ to *n*-C_35_), which reflects the distinction in *n*-alkane profiles observed between olive leaves and olives.

#### 2.3.4. Influence of Oil Refining

According to some authors [25], based on *n*-alkanes composition (quali-/quantitative profile), it is possible to distinguish between crude and refined oils. Lanzon et al. [25] analyzed virgin olive oils obtained by physical refining and by conventional alkaline neutralization and steam deodorization. They found that the refined oil differed from the corresponding virgin olive oil, both from a qualitative and a quantitative point of view. Compared to oils obtained with conventional refining, greater differences were found in oils subjected to the more drastic physical refining. Physical refining leads to the loss of lower-molecular-weight *n*-alkanes and, as a consequence, to higher *n*-alkane content in virgin olive oils than in the corresponding refined oils. Total *n*-alkanes changed from 27.7 to 20.9 mg/kg and from 21.8 to 14.69 mg/kg, for conventional and physical refining, respectively.

Webster et al. [5] analyzed samples of virgin and refined olive oils to evaluate if *n*-alkane profile and composition were oil-specific and, therefore, if it could be used for evaluating olive oil adulteration. The total *n*-alkane concentration in authentic olive oils varied from 18.6 mg/kg for a Spanish olive oil to 175.7 mg/kg for a Greek sample. A similar value range was found in refined olive oils, where total *n*-alkanes ranged from 15.0 mg/kg for a Spanish olive oil to 174.3 mg/kg for a Tunisian one. Nevertheless, when the mean values were compared, EVOO presented higher total concentration (71.6 mg/kg) with respect to the refined ones (36.2 mg/kg) (Table 1), probably due to the refining process, which reduces the concentration of the shorter chain length *n*-alkanes. 

Benitez-Sánchez et al. [29] studied the composition of crude and refined hazelnut oils from different countries and found that, compared with crude oils, refined oils had half the *n*-alkane content (See Table 2).

Srbinovska [32] reported important differences in the *n*-alkane content of crude and refined sunflower and avocado oils, with higher amount in the crude oil (83.9 and 54.7 mg/kg for crude and refined sunflower oil, respectively; 317.9 and 167.0 mg/kg for crude and refined avocado oil, respectively).

## 3. Analytical Determination

### 3.1. Methods Specifically Designed for n-Alkane Determination in Vegetable Oils

Starting from the 1990s, relatively few authors focused on analytical determination of *n*-alkanes in vegetable oils. Table 3 and Table 4 summarize these works, reporting the main steps involved in the sample preparation and analytical determination. More precisely, Table 3 refers to methods involving a saponification step, while Table 4 refers to methods involving a passage on a fat retainer followed by analytical determination.

Most of the methods used for *n*-alkane determination involve long sample preparation steps and high solvent consumption. Nevertheless, as later discussed, *n*-alkane determination can be more easily accomplished by using methods developed for mineral oil analysis, in particular for the saturated fraction called MOSH, which will be discussed in Section 3.3.

Capillary GC, coupled with an FID, which provides virtually equal response per unit of mass for all hydrocarbons, represents the most suitable method for analytical determination of *n*-alkanes. Nevertheless, some authors who extended their research on other hydrocarbon classes (alkenes, sesquiterpenes, etc.) used MS as an additional detector for identification purpose [2,25,26]. GC can separate practically all hydrocarbons, and their mass spectra are easily distinguished from other lipid components [25,33]. Troya et al. [3] used *n*-alkane profiles established by GC–MS to classify vegetable oils according to their botanical origin. Mihailova et al. [12], investigated stable carbon and hydrogen isotope analyses of *n*-alkanes by using an isotope ratio mass spectrometer interfaced with GC.

Before GC analysis, a purification step must be applied to eliminate the fat. A first approach used to eliminate triglycerides is saponification, followed by unsaponifiable extraction and fractionation on a glass column filled with silica gel (Table 3).

The saponification step is generally performed [1,25,27,30] according to the official method of the European Community (Annex XVII, 1995), designed for the determination of stigmastadienes in vegetable oils [35]. Some authors introduced little modifications to this protocol, aimed at speeding up unsaponifiable extraction by reducing the number of washing in separatory funnels [2] and/or at modifying the saponification condition [3,23].

After saponification, the *n*-alkane fraction needs to be separated from the rest of unsaponifiable matter. *n*-Alkanes are apolar compounds that, when loaded on a polar column (silica), elute soon after the dead volume. Table 3 reports the different elution conditions used by different authors. In general, a glass column is filled with silica gel (slurried in hexane), loaded with the unsaponifiable fraction and eluted with 50–120 mL of hexane to collect the *n*-alkane fraction.

Instead of using column chromatography, Sakouhi et al. [23], used thin-layer chromatography (TLC) to isolate the *n*-alkane fraction from the unsaponifiable matter.

Alternatively, as summarized in Table 4, to avoid saponification, *n-*alkanes can be directly separated from the lipid extract using column chromatography on a fat retainer [5,12,24,28]. To isolate the *n*-alkane fraction of edible oils, McGill et al. [24] and Webster et al. [5] used a glass column (11 × 2 cm i.d.) packed with silica gel heated at 550 °C for 18 h and deactivated with 1% of water. After sample loading, the hydrocarbon fraction was eluted with 100 mL of hexane [24] or 130 mL of isohexane [5]. In both cases, the isolation of the *n*-alkane fraction from other aromatic compounds was accomplished by a silica HPLC column (Lichrosorb Si-60, 5 µm) using hexane [24] or isohexane [5] as eluent.

Benitez-Sánchez et al. [29] performed sample preparation according to the ISO method described in Section 3.2., which again requires high solvent volumes for sample processing.

Mihailova et al. [12] replaced the glass column with a Pasteur pipette filled with silica gel, thus reducing by far the amount of sorbent used as fat retainer and hence the volume of the elution solvent (5 mL of hexane). Nevertheless, to avoid excessive loss of sensitivity, each oil sample was extracted twice, and the eluents were combined and concentrated to 0.5 mL before subsequent GC analysis (splitless mode).

Finally, the hydrocarbon fraction can be analyzed by the online HPLC-GC-FID method. An online method proposed by Moret et al. [36] for mineral paraffins was successfully used in variety differentiation of virgin olive oil based on *n*-alkane profile [11].

In contrast to traditional methods involving the saponification and/or passage on large packed silica gel column, online methods allow one to perform rapid automated analysis, reducing the solvent consumption and time for sample preparation and obtaining optimal reproducibility characteristics. 

Concerning the method performance, only a few authors validated the method used for *n*-alkane determination. Among these, Koprivnjak et al. [27], who performed saponification followed by unsaponifiable extraction, fractionation on silica gel and GC-FID analysis, evaluated method performance by recovery and repeatability tests. For this purpose, a soybean oil with low *n*-alkane content was spiked with three standards (C_17:1_, C_26_, C_36_) covering the start, the middle and the end of the molecular range of the aliphatic hydrocarbons present in virgin olive oils. Recovery ranging from 91 to 101% was found. Repeatability tests (10 replicates) on an extra virgin olive oil showed for 15 components an average coefficient of variation of 2.2%. Higher values (but lower than 7%) were found for components present at very low concentrations.

Bortolomeazzi et al. [2], who used a similar method followed by GC-MS, evaluated for each of the *n*-alkanes the linearity of the detector response, finding correlation coefficients higher than 0.9991. When using automated integration, the limit of detection (LOD) of *n*-alkanes was 25 pg (in the column). Lower LOD (5 pg) was reached when using manual integration. Recoveries (six replicates), determined by extracting a known amount of a standard mixture containing *n*-C_13_, *n*-C_21_ and *n*-C_29_, using the same procedure as for edible oils, were around 86% for *n*-C_13_ (with residual standard deviation of 11.2%) and approaching 100% for the remaining components.

### 3.2. Official Method for the Determination of Aliphatic Hydrocarbons in Vegetable Oils 

In 2015, the reference method (ISO WD 17780:2015) [18,37] for the determination of saturated aliphatic hydrocarbons from *n*-C_10_ to *n*-C_56_ in vegetable fats and oils was published. This International Standard, which can be used for the analysis of saturated aliphatic hydrocarbons of natural origin present in the vegetable oils, as well as for detecting the presence of mineral oil and diesel oil, is based on the following principle. The saturated aliphatic hydrocarbons are isolated by liquid chromatography on silica gel impregnated with silver nitrate and determined by GC-FID, using *n*-C_18_ as internal standard. A glass column is filled with 18.5 g of silver silica (10%) surmounted by a 0.5–1 cm layer of sodium sulfate. The method allows withstanding the loading of 1 g of sample, but with a significant solvent consumption: 40 mL of *n*-hexane to pack the column, 60 mL to condition it, and then 55 mL to elute the *n*-alkane fraction. After solvent evaporation to 0.5 mL, 2 µL is injected on the column for quantification.

This standard also reports the possibility of using a quick method for refined or virgin/cold-pressed oils. This variant involves loading 250 mg of oil sample onto a cartridge directly packed with 2 g of non-activated silica and elution with 3.5 mL of *n*-hexane. According to the method, batches of silica with low activity (resulting in poor chromatographic separations) need to be activated by heating (4 h at 500 °C) and water addition (2%). In this case, the sample was not concentrated, and 45 µL was injected directly into the GC-FID in the large volume mode. Due to the possible lack of retention towards triglycerides, this quick method is not suitable for crude oil.

### 3.3. Methods Developed for Mineral Oil Analysis

As already mentioned, in practice all the methods developed for mineral oil analysis, except those designed to eliminate *n*-alkanes (which may give interference when co-eluting with mineral oils), can be used for evaluating endogenous *n*-alkanes in vegetable oils. A number of reviews described methods for mineral oil determination in vegetable oils [38,39,40], so readers can refer to them for more detailed information.

Online HPLC-GC, which is the reference method for mineral oil determination in vegetable oils and fat [13,41] is also the best method for rapid and solvent sparing *n*-alkane determination. In this case the glass column is replaced by an HPLC silica column able to retain the fat, letting the aliphatic hydrocarbons elute soon after the dead volume. The fraction of interest is then transferred to the GC. Over the years, different interfaces, equipped with an exit for solvent vapor discharge have been proposed [42,43]. During the GC analysis, the LC column is backflushed to eliminate the fat and reconditioned before the subsequent analysis.

Besides online methods, which require expensive apparatus and skilled operators, in recent years, different off-line solid-phase extraction (SPE)-GC methods have been developed for mineral oil determination in vegetable oils. In this section, the focus will be on methods involving reduced solvent consumption, which could be successfully used for *n*-alkane determination.

A first method reported by Fiorini et al. [21] enabled the loading of 150 mg of the oil sample (dissolved in 200 µL of *n*-hexane) onto a 2 g column of non-activated silica gel. The MOSH fraction was eluted with 5 mL of *n*-hexane, and, after concentration to 300 µL, the sample was injected into the GC-FID (splitless injection of 1 µL). The use of non-activated silica gel reduces the retentive power towards the saturated hydrocarbons, reducing the volume of the elution solvent, but also the triglycerides are less retained, limiting the amount of oil that can be loaded.

The retentive properties of different adsorbents and their activation degree were also evaluated by Moret et al. [15]. The use of silver silica (10%) was chosen for its reduced retention towards saturated hydrocarbons and enhanced retention towards squalene and aromatics. A glass cartridge filled with 1 g of 10% silver silica and conditioned with 5 mL of *n*-hexane was loaded with 125 mg of oil (diluted to 500 µL). The MOSH fraction, and hence *n*-alkanes, eluted with only 1.5 mL of *n*-hexane, after discharging the dead volume (1 mL). Of the collected fraction, 40 µL were then injected large volume into the GC-FID. 

A similar method was proposed by Liu et al. [19]. Briefly, a glass SPE cartridge filled with 2 g of activated silica gel coated with 1% silver nitrate was loaded with 0.2 g of oil, whose MOSH fraction was collected in 3 mL of *n*-hexane, after discarding 1 mL of dead volume. Fifteen microliters of the final solution were injected large volume, after reconcentration to 0.5 mL. 

## 4. Conclusions

From the various works examined in the present review, it emerged that different types of vegetable oils show a significantly different *n*-alkane pattern. Moreover, different *n*-alkane profiles can also be found when analyzing the same type of oil if obtained from fruits of different provenance, cultivar or maturation degree or subjected to different refining processes. Although works focused on analytical determination of *n*-alkanes are relatively few, it is evident how this parameter could be exploited to characterize different oils in order to determine their authenticity and to highlight the presence of adulterations. Furthermore, the application of chemometrics could help in developing a powerful tool for quality control of edible oils.

The majority of the analytical methods reported in the literature to quantify *n*-alkanes, comprising the ISO official standard for the determination of saturated aliphatic hydrocarbons in vegetable fats and oils, are solvent- and time-consuming. Rapid and solvent sparing methods, based on online HPLC-GC-FID or off-line SPE-GC-FID, could be advantageously used, after validation, by significantly simplifying the analysis.

## Figures and Tables

**Figure 1 foods-09-01546-f001:**
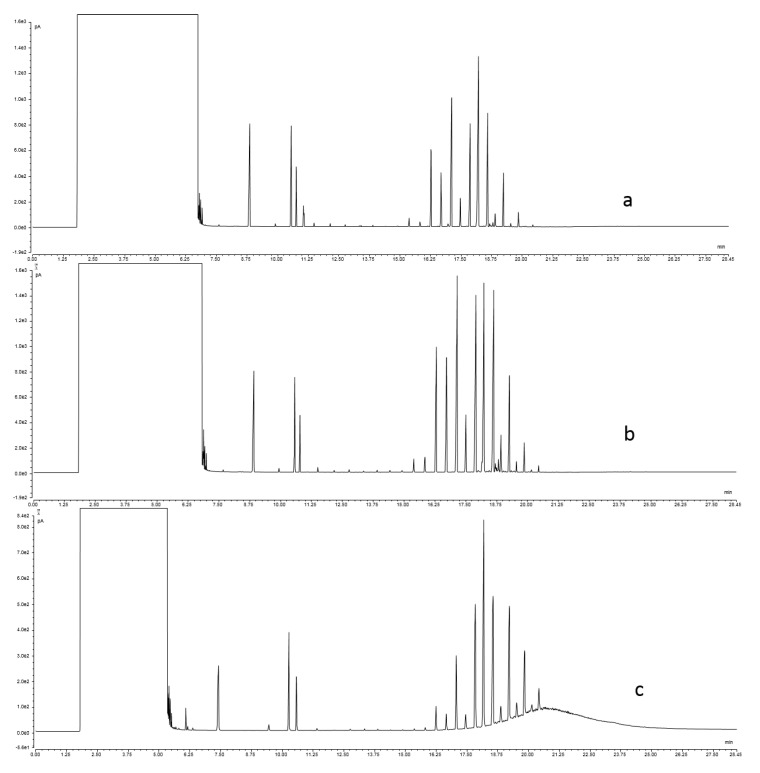
(**a**) Bianchera and (**b**) Leccino—Typical HPLC-GC-FID traces of extra virgin olive oils without MOSH presence; (**c**)—olive oil trace with mineral oil hydrocarbons (MOSH) presence.

**Figure 2 foods-09-01546-f002:**
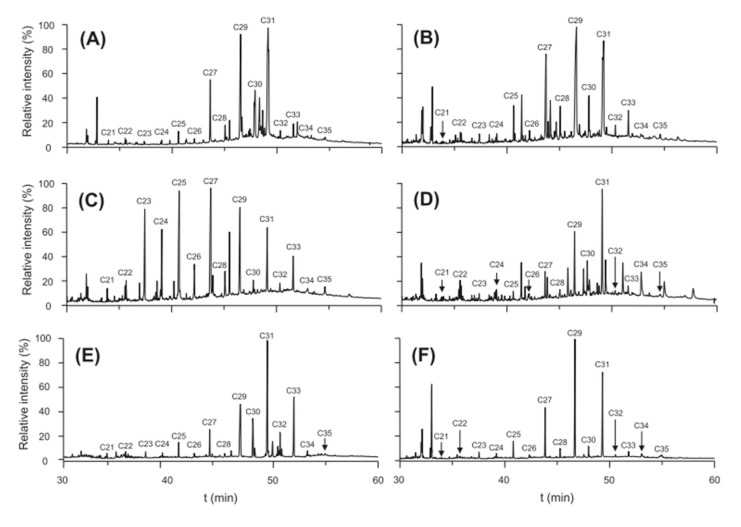
GC-MS chromatograms of different vegetable oils: (**A**) grapeseed oil, (**B**) hazelnut oil, (**C**) olive oil (Arbequina), (**D**) peanut oil, (**E**) corn oil and (**F**) sunflower. Reference [3] with permission from Elsevier.

**Figure 3 foods-09-01546-f003:**
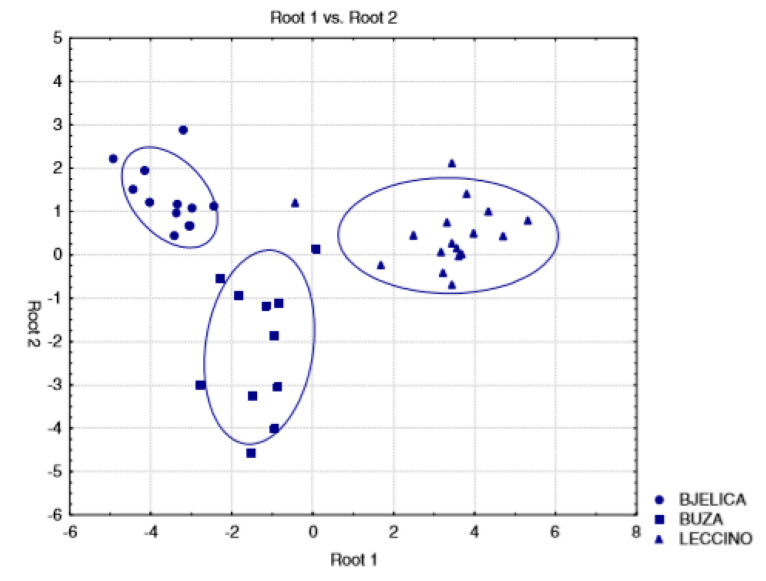
Discriminant analysis—scatterplot of canonical scores (range, coefficient 0.95). Reference [11] with permission from Elsevier.

**Figure 4 foods-09-01546-f004:**
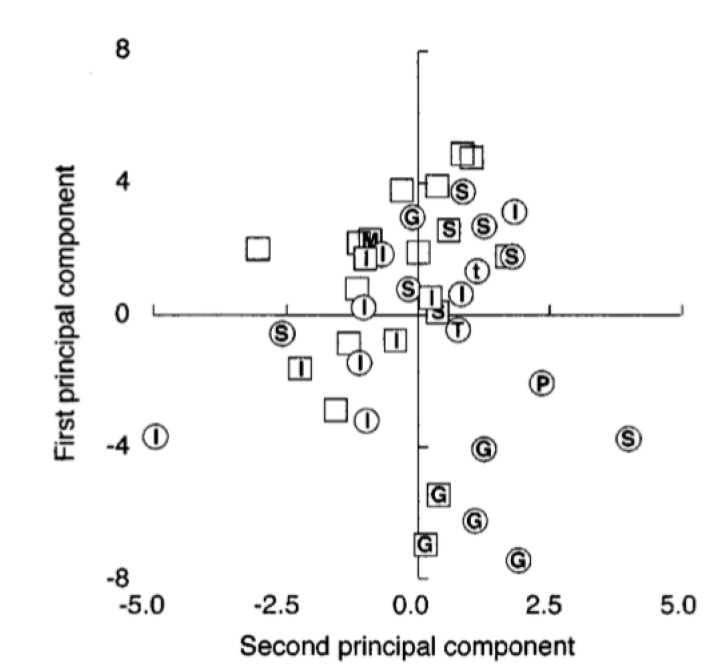
PCA results of the n-alkanes (*n*-C_15_-*n*-C_35_) in authentic (○ ) and commercial (□) extra olive oils. By plotting the first principal component against the second principal component, the Greek oil (G) is shown to be well resolved from Spanish (S) and Italian (I) oils. Reference [5] with permission from The Royal Society of Chemistry.

**Table 1 foods-09-01546-t001:** Summary of n-alkanes presence in extra virgin olive oils, olive oils and refined olive oils.

Sample	Variety	Geographical Origin	Sample Number	Tot. *n*-Alkane (mg/kg)	Major *n*-Alkanes	Ref.
Olive oil and Extra virgin olive oil			6	28–99	*n*-C_23_, *n*-C_25_, *n*-C_27_	[24]
Virgin olive oil	Arbequina, Cornicabra, Empeltre, Farga, Hojiblanca, Lechin, Picual, Picudo, Verdial	Various Spanish regions	250	15.3–115.8	*n*-C_25_, *n*-C_23_, *n*-C_27_	[25]
Refined virgin olive oil			2	20.9	*n*-C_27_, *n-*C_29_, *n*-C_31_	
				14.7	*n*-C_29_, *n*-C_27_, *n*-C_31_	
Virgin olive oil	Arbequina	Lèrida	50	36–60	*n*-C_27_, *n*-C_25_, *n*-C_29_	[26]
	Carnicabra	Toledo		26–43	*n-*C_29_, *n*-C_27_, *n*-C_31_	
	Empeltre	Teruel		72–92	*n*-C_25_, *n*-C_23_, *n*-C_29_	
	Hojiblanca	Màlaga/Còrdoba		20–53	*n*-C_27_, *n*-C_29_, *n*-C_25_	
	Picual	Jaen		18–31	*n-*C_27_, *n*-C_29_, *n*-C_25_	
Virgin olive oil	Debela	Krk island (Croatia)		31	*n*-C_29_, *n*-C_27_, *n*-C_31_	[27]
	Naska			104	*n-*C_25_, *n*-C_23_, *n*-C_24_	
	Rosulja			42	*n-*C_29_, *n*-C_25_, *n*-C_27_	
	Slatka			40	*n*-C_29_, *n*-C_27_, *n*-C_25_	
Extra virgin olive oil		Various origin	20	71.6		[5]
Refined olive oil		Various origin	20	36.2		
Virgin olive oil		Various origin	30	30.5–176.2	*n*-C_23_, *n-*C_25_, *n*-C_27_, *n*-C_29_	[2]
Virgin olive oil	Bjelica	Istria (Croatia)	40	13.7−26.4	*n-*C_29_, *n*-C_25_, *n*-C_27_	[11]
	Buza			28.0−62.3	*n*-C_25_, *n*-C_23_, *n*-C_24_	
	Leccino			27.0− 80.4	*n*-C_25_, *n-*C_23_, *n*-C_24_	
Olive oil	Meski	Bzerte (North-East Tunisia)		90.9 ± 22.7	*n*-C_25_, *n*-C_27_, *n*-C_23_	[23]
Virgin olive oil	Frantoio, Leccino, Maraiolo	Ciggiano, Italia	8		At early ripening *n*-C_29_ and *n-*C_31_;At late ripening *n*-C_25_, *n*-C_27_, *n*-C_29_	[28]
Olive oil			4		*n-*C_27_, *n*-C_25_, *n*-C_29_	[3]
Extra virgin olive oil	Bianchera istriana, Leccino, Frantoio	Koper (Slovenia)	3	62.7 ± 4.5	*n-*C_29_, *n*-C_25_, *n*-C_27_	[12]
	Buza	Vodnjan (Croatia)	4	96.7 ± 12.8	*n*-C_25_, *n*-C_23_, *n*-C_29_	
	Salonenque, Aglandau, Grossane, Verdale, Altre, Calian, Petit ribier, Coratina	Provence-Alpes-Côte d’Azur (France)	15	66.8 ± 10.6	*n*-C_25_, *n-*C_29_, *n*-C_27_	
	Leccino, Frantoio, Grignano, Perlarola	Veneto (North Italy)	4	81.3 ± 13.3	*n*-C_25_, *n*-C_27_, *n-*C_23_	
	Frantoio, Moraiolo, Leccino, San Felice, Caninese, Itrana	Tuscany, Laizo, Umbria (Central Italy)	14	115.6 ± 12.1	*n*-C_25_, *n*-C_29_, *n*-C_27_	
	Ogliarola Precoce, Nocellara del Belice, Biancolilla, Tonda Iblea	Apulia (South Italy)	7	96.2 ± 22.0	*n*-C_25_, *n*-C_23_, *n*-C_29_	
	Cobrancosa, Madural, Arbequina, Verdeal transmontana, Galega vulgar	Portugal	5	73.9 ± 12.5	*n*-C_29_, *n-*C_27_, *n-*C_25_	
	Chalkidiki	Central Macedonia (North Greece)	8	35.6 ± 3.5	*n*-C_25_, *n*-C_27_, *n*-C_29_	
	Koroneiki	Peloponnese (South Greece)	6	67.8 ± 5.2	*n-*C_23_, *n*-C_25_, *n*-C_27_	
	Picual	Jean (Spain)	6	40.1 ± 7.1	*n-*C_29_, *n*-C_27_, *n-*C_25_	
	Picholine marocaine	Meknes, El Hajeb (Marocco)	4	78.0 ± 11.6	*n*-C_23_, *n*-C_25_, *n*-C_27_	

**Table 2 foods-09-01546-t002:** Summary of *n*-alkane presence in various vegetable oils.

Sample	Variety	Geographical Origin	Sample Number	Tot. *n*-Alkane (mg/kg)	Major *n*-Alkanes	Ref.
Sunflower oil			5	105–166	*n*-C_27_, *n-*C_29_, *n*-C_31_	[24]
Sesame oil			4	7–30	*n*-C_29_, *n*-C_31_, *n*-C_27_	
Corn oil			3	26–33	*n*-C_31_, *n*-C_29_, *n*-C_27_	
Walnut oil			3	7–30	*n*-C_29_, *n*-C_27_, *n*-C_31_	
Peanut/Groundnut oil			3	27–40	*n*-C_29_, *n*-C_31_, *n*-C_27_	
Hazelnut oil (crude)		Turkey, France, Belgium, Italy, Spain	10	0.6–13.9	*n-*C_25_, *n*-C_23_, *n*-C_27_	[29]
Hazelnut oil (refined)			7	0.6–5.17	*n*-C_29_, *n*-C_25_, *n*-C_27_	
Avocado oil	Persea americana	Gioia Tauro (Italy)		16.8–25.2	*n*-C_24_, *n*-C_25_, *n*-C_23_	[30]
Grapessed oil			4		*n-*C_31_*, n-*C_29_, *n*-C_27_	[3]
Hazelnut oil			4		*n-*C_29_, *n*-C_31_, *n*-C_27_	
Peanut oil			4		*n*-C_31_, *n*-C_29_, *n*-C_27_	
Corn oil			4		*n*-C_31_, *n*-C_33_, *n*-C_29_	
Sunflower oil			4		*n*-C_29_, *n*-C_31_, *n*-C_27_	
Tomato seed oil	Principe Borghese	Southern Italy		347.3 ± 4.04	*n*-C_21_, *n*-C_25_, *n*-C_23_	[1]
	Rebelion			216.0 ± 3.05		
	San Marzano			106.7 ± 0.58		

**Table 3 foods-09-01546-t003:** Methods based on saponification for n-alkane determination in vegetable oils.

Sample Amount—Internal Standard (I.S.)	Saponification and LLE for Unsaponifiable Extraction	Column Chromatography and/or Further Purification	Analysis (Injection Mode)—Column Type—Chromatographic Conditions	Ref.
20 g oil sample + I.S. (*n*-C_20_) in 1 mL hexane	According to the Official European Community method—Annex XVII, for the determination of stigmastadienes in vegetable oils, 1995 *.	A glass column (1.5 cm i.d. × 50 cm) was filled with 15 g silica gel, slurried in hexane. After sample loading, the *n*-alkane fraction was eluted with hexane (50 mL) at 1 mL/min and concentrated to 1 mL.	GC-FID/ GC-MS Wide bore capillary column of borosilicate glass (30 m × 0.75 mm i.d.), coated with SPB-1, 1 µm film thickness. Oven temperature: 110 °C (6 min) to 300 °C at 5 °C/min. Injector: 300 °C; Detector: 320 °C; carrier gas: nitrogen at 14 mL/min.	[25]
20 g oil sample + I.S. (*n-C_20_*) in 1 mL hexane	According to the Official European Community method—Annex XVII, for the determination of stigmastadienes in vegetable oils, 1995 *.	A glass column (2.0 cm i.d. × 50 cm) was filled with 15 g silica gel slurried in hexane. After sample loading, the *n*-alkane fraction was eluted with hexane (50 mL) and concentrated.	GC-FID (split/splitless injection; split ratio 1:20). Fused silica capillary column (30 m × 0.32 mm i.d.), coated with SPB 5; 1.0 µm film thickness. Oven temperature: 120 °C (4 min) to 280 °C (5 min) at 4 °C/min, then to 305 °C (10 min) at 4 °C/min; Injector and detector: 315 °C; carrier gas: helium (2 mL/min).	[27]
20 g oil sample + I.S. *(n*-C_11_)	According to the Official European Community method—Annex XVII, for the determination of stigmastadienes in vegetable oils, 1995 *.	A glass column (1.5 cm i.d. × 50 cm) was filled with a 2 mm layer of anhydrous sodium sulphate and 15 g of silica gel and conditioned with hexane. After sample loading, the *n*-alkane fraction was eluted with *n*-hexane (120 mL) and concentrated to 1 mL.	GC-FID (on-column injection). Fused silica capillary column (25 m × 0.32 mm i.d.), coated with SE54, 0.25 µm film thickness. Oven temperature: 60 °C (1 min) to 290 °C (40 min) at 5 °C/min. Detector: 310 °C.	[30]
20 g oil sample + I.S. (*n*-C_20_) in 1 mL hexane	According to the Official European Community method—Annex XVII, for the determination of stigmastadienes in vegetable oils, 1995 *. The solution was then passed through anhydrous sodium sulphate (50 g), washed with 20 mL of hexane and evaporated to dryness.	A glass column was filled with silica gel. After sample loading, the *n*-alkane fraction was eluted with hexane (30 mL).	GC-FID (split injection). Fused silica capillary column (30 m × 0.32 mm i.d.), coated with SE 54, 0.5 µm film thickness. Oven temperature: 100 °C (1 min) to 300 °C (10 min) at 4 °C/min. Injector: 280 °C; detector: 310 °C; carrier gas: helium (10 psi).	[1]
20 g oil sample + I.S. (*n*-C_20_) in 1 mL hexane	Saponification: 30 min slight boiling with 75 mL of 10% ethanolic KOH. LLE: the saponified solution was transferred to a 500 mL separatory funnel, added to 100 mL of distilled water and extracted twice with 100 mL hexane. The combined extracts were concentrated to 1 mL.	A glass column (1.5 cm i.d. × 50 cm) was filled with 15 g silica gel, slurried in hexane. After loading the unsaponifiable fraction (together with washes), the *n*-alkane fraction was eluted with hexane (60 mL) at a rate of about 1 mL/min.	GC-FID/ GC-MS. Wide bore capillary column of borosilicate glass (30 m × 0.75 mm i.d.), coated with SPB-1, 1 µm film thickness. Oven temperature: 110 °C (60 min) to 300 °C at 5 °C/min. Injector: 300 °C; detector: 320 °C; carrier gas: nitrogen (14 mL/min).	[26]
20 g of oil sample + I.S. (*n*-C_20_) in 1 mL hexane	Saponification 30 min slight boiling with 75 mL of 10% ethanolic KOH. LLE: The saponified solution was transferred to a 500 mL decanting funnel, 100 mL distilled water was added, and the mixture was extracted twice with 100 mL portions of hexane. The hexane solution was dried over anhydrous sodium sulfate, evaporated to dryness and dissolved in 1 mL of hexane	A glass column (1.5 cm i.d. × 40 cm) was filled with 15 g of silica gel. After sample loading, the *n*-alkane fraction was eluted with hexane (100 mL) and concentrated to about 0.5 mL.	GC-MS (on column injection). Fused silica capillary column (30 m × 0.25 mm i.d.), coated with DB-5, 0.25 µm film thickness. Oven temperature: 60 °C (1 min) to 120 °C at 3 °C/min, and finally to 300 °C at 7 °C/min. Detector: 300 °C; carrier gas: helium (1.3 mL/min).	[2]
5 g of oil sample + I.S. (*n*-C_16_)	Saponification: ethanolic KOH 12% (*w*/*v*) at 60 °C for 1.5 h. After cooling, 50 mL of water was added and the unsaponifiable matter was extracted with 4 × 50 mL of petroleum ether. The combined extracts were washed with 50 mL of ethanol: water (1:1), dried over anhydrous sodium sulfate and evaporated to dryness.	The dry residue was dissolved in chloroform and deposited on a silica gel TLC plate. After developing the plate with hexane/diethyl ether (6:4, *v*/*v*), it was sprayed with 2,7-dichlorofluorescein and the band corresponding to hydrocarbons was scraped, extracted 3 times with chloroform/diethyl ether (1:1, *v*/*v*), filtered and dried in a rotary. evaporator.	GC-FID (split injection; 60:1 split ratio). Fused silica capillary column (30 m × 0.25 mm i.d.), coated with DB-5MS, 0.25 µm film thickness. Oven temperature: 150 °C to 300 °C (10 min) at 4 °C/min. Injector and detector: 250 °C.; carrier gas: helium (mL/min). The transfer line: 250 °C. Electron impact mass spectra were measured at an acceleration energy of 70 eV.	[23]
5 g of oil sample + I.S. *(n*-C_15_)	Saponification: 19 mL of a 10% ethanolic KOH under reflux for 20 min. After cooling, 25 mL distilled water was added and the unsaponifiable fraction was extracted with 2 × 25 mL of *n*-hexane. The combined extracts were washed with 3 × 12.5 mL of ethanol:water mixture (1:1), dried on anhydrous sodium sulfate and evaporated to dryness.	A glass column (1.5 cm i.d. × 40 cm) was filled with 15 g of silica gel. After sample loading, the *n*-alkane fraction was eluted with *n*-hexane (40 mL) and concentrated to 0.5 mL.	GC-MS (splitless injection)—Fused silica capillary column (30 m × 0.25 mm i.d.), coated with TR-5MS, 0.25 µm film thickness. Oven temperature: 60 °C (3 min) to 300 °C (10 min) at 5 °C/min. Injector and transfer line at 300 °C. Ion source temperature: 225 °C; Carrier gas: helium (1 mL/min).	[3]

* According to the official method of the European Community (13)—Annex XVII, designed for the determination of stigmastadienes in vegetable oils (ECC, 1995): Saponification: 30 min slight boiling with 75 mL of 10% etanolic KOH. LLE: the saponified solution was transferred to a 500 mL separatory funnel, added to 100 mL of distilled water and extracted twice with 100 mL hexane; the combined extracts were washed with a mixture of EtOH:H_2_O (1:1) until reaching neutral pH. The unsaponifiable fraction was dried over anhydrous sodium sulfate, evaporated to dryness and dissolved in 1 mL of hexane. LLE, liquid–liquid extraction.

**Table 4 foods-09-01546-t004:** Methods based on use of fat retainers for n-alkane determination in vegetable oils.

Sample Amount—Internal Standard (I.S.)	Column Chromatography	Analysis (Injection Mode)—Column Type—Chromatographic Conditions	Ref.
100 mg oil sample in 5 mL hexane + I.S. mixture	A glass column (46 cm^3^) filled with silica gel heated al 550 °C for 18 h, pre-washed with 100 mL of hexane. After the sample was loaded, the *n*-alkane fraction was eluted with hexane (100 mL) and concentrated to 150 µL, followed by HPLC separation from aromatics on a 25 × 0.46 cm Lichrosorb Si-60, 5 µm.	GC-FID (on-column injection). Fused silica capillary column (25 cm × 0.25 mm i.d.), coated with 0.33 µm film of Ultra 1. Oven temperature: 60 °C (3 min) to 280 °C at 4 °C/min and held at this temperature until the end of the run. Injector and detector: 300 °C; carrier gas: nitrogen.	[24]
500 mg of oil sample + I.S. mixture in 5 mL of isohexane	A glass column (2.0 cm i.d. × 11cm) filled with silica gel heated at 550 °C for 18 h and deactivated with 1% water, was washed with iso-hexane (100 mL). After loading the sample, the *n*-alkane fraction was eluted with isohexane (130 mL) and concentrated to 0.3 mL. Followed HPLC separation from aromatics on a 25 × 0.46 cm Lichrosorb Si-60, 5 µm.	GC-FID (on-column injection). Fused silica capillary column (25 cm × 0.25 mm i.d.), coated with Ultra 1, 0.33 µm film thickness. Oven temperature: 60 °C (3 min) to 280 °C at 4 °C/min and held at this temperature until the end of the run. Injector and detector: 300 °C; carrier gas: nitrogen.	[5]
60 mg of oil sample + I.S. *(n*-C_13_, C_14:1_) in pentane	Online LC-GC-FID.	LC column: silica (Spherisorb), 100 × 4.6 mm i.d; wire interface; GC column: fused silica capillary column (12 m × 0.25 mm i.d.), coated with immobilized PS-255, 0.3 µm of film thickness. Transfer occurred at 49 °C; 8 min after the transfer the temperature was programmed at 15 °C/ min to 320 °C. The carrier gas (helium) inlet pressure was 70 kPa.	[11]
75 mg of oil sample + I.S. (*n*-C_15_) in 0.5 mL of hexane	A glass Pasteur pipette (145 mm × 9 mm, packed with activated silica gel, was loaded with the sample and eluted with 5 mL of hexane. Each oil sample was extracted twice, and the eluents were combined and concentrated to 0.5 mL.	GC-FID (splitless mode) GC. Fused silica capillary column (30 m × 0.32 mm), coated with DB-5, 0.25 µm film thickness. Oven temperature: 50 °C (1 min) to 150 °C at 20 °C/min, then to 230 °C (10 min) at 8 °C/min. Injector: 300 °C; detector 280 °C; carrier gas: helium (1.2 mL/min). As an additional tool for [12], stable carbon and hydrogen isotope analyses of *n*-alkanes were performed using an isotope ratio mass spectrometer interfaced with GC.	[12,28]
1 g of oil sample + I.S. (*n*-C_15_)	A glass column (1.5 cm i.d. × 50 cm) was filled with 40 mL hexane. About 18.5 g of silver-silica gel was poured slowly, avoiding bubbles. Later, a 1 cm layer of anhydrous sodium sulfate followed by a 1 cm layer of washed sand was added. Finally, the remaining solvent above the sand was discarded. After the sample was loaded, saturated hydrocarbons were eluted with 55 mL of hexane, concentrated and re-dissolved in 1 mL of heptane.	GC-FID (PTV). Fused silica capillary column (10 m × 0.32 mm) coated with CP-9070, 0.10 µm film thickness. Oven temperature: 60 °C (1 min) to 288 °C at 12 °C/min, then to 340 °C (4 min) at 6 °C/min. Detector temperature: 350 °C. Injection temperature program: 70 °C (1 min) to 300 °C (1 min) at 200 °C/min, and then back to 70 °C. Carrier gas: hydrogen at 15 mL/min.	[34]

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
