# Peer review of "Occurrence of n-Alkanes in Vegetable Oils and Their Analytical Determination"

_foods, 2020, doi:10.3390/foods9111546_

Round 1

Reviewer 1 Report

The submited review gives a good survey on the occurrence of n-alkanes in vegetable oils - with an emphasise on olive oil - and their analytical determination. The results of cited works of the last ~30 years demonstrate,  that different types of vegetable oils -and even the same type of oil - show significantly different n-alkane patterns, depending on various factors, which are discussed in detail. On the other side, these patterns and its analysis can be a worthful tool to  characterize different oils in order to determine their authenticity and the presence of adulterations. Most common analytical methods are described and its application and limits are  discussed.

The article is well organized, well readable and understandable. Although not a native speaker, my impression is that English grammar and style needs at least moderate changes. I have indicated several hints and proposals which should improve readability and clearness directly  in my corrected version.

My main concern is, that a lot of published papers in the treated field is missing. I am missing several preceding reviews and original papers close to your topics, e.g.:

- Reviews: Ramón Aparicioa , Ramón Aparicio-Ruı́zb : Authentication of vegetable oils by chromatographic techniques; Journal of Chromatography A, Volume 881, Issues 1–2, 9 June 2000, Pages 93-104

- Aluyor, E. O.1*, Ozigagu, C. E.1, Oboh, O. I.2 and Aluyor, P.: Chromatographic analysis of vegetable oils: A review; https://academicjournals.org/article/article1380791937_Aluyor%20et%20al.pdf

Missing papers:

- Magdalena LigorBogusław Buszewski: The comparison of solid phase microextraction-GC and static headspace-GC for determination of solvent residues in vegetable oils, J. Sep. Sci. 2008,31, 364–371

- Raquel B. Gómez-Coca, María del Carmen Pérez-Camino & Wenceslao Moreda: Saturated hydrocarbon content in olive fruits and crude olive pomace oils, https://doi.org/10.1080/19440049.2015.1133934

- same authors: Determination of saturated aliphatic hydrocarbons in vegetable oils; http://dx.doi.org/10.3989/gya.0627152

- F. Lacoste: International validation of the determination of saturated hydrocarbon mineral oil in vegetable oils; https://doi.org/10.1002/ejlt.201500134

Some additional information and references on coupled techniques using headspace and/or SPME with HPLC and /or MS could also be found in scientific literature and could improve your article.

All in all the paper is of interest for institutes and authorities in food control as well as for oil producers.

Reviewer 2 Report

Occurrence of n-alkanes in vegetable oils and their analytical determination. Srbinovska, Conchione, Ursol, Lucci and Moret.

 An extensive review of the literature on the occurrence of n-alkanes in vegetable oils, with a focus on olive oil. The review identifies strengths and weaknesses in the data and highlights potential applications.

The manuscript is an interesting read, is well written and well organized, but would benefit from a little polishing of the English. For example:

Line 11. “odd terms prevalence” might read better as “odd chain lengths prevalent”.

Line 15. Remove the word “instead”.

Line 31. Perhaps add the word other before “ living organisms”

Line 52. A stray sentences that doesn’t really deserve to be separate paragraphs.

There are a few points that it would be nice to see addressed.

A bit more discussion on what else is in the non-saponifiable fraction of vegetable oils. (alkanes are not the most abundant components)

Briefly discus the merits of alkane profiling compared to other markers, such as sterols and triterpenoids, for product identification and detection of adulterants.

By focusing on n-alkanes researchers may have overlooked branch chain components (iso and anteiso for example). Is there any literature on this?

Minor points.

A good review to reference for plant surface wax is Samuels et al 2008. (Samuels L, Kunst L, Jetter R. 2008 Annu Rev. Plant Biol. 59:683-707. https://doi.org/10.1146/annurev.arplant.59.103006.093219

This would cover surface composition and also alkane biosynthesis (a reference to use for Line 39).

Lines 32 and 33, add triterpenoids.
